# Analysis of the Resistance Change of Chemosensitive Layers to the Presence of Ammonia Vapors under Variable Conditions of Air Temperature and Humidity

**DOI:** 10.3390/polym15020420

**Published:** 2023-01-13

**Authors:** Hanna Zajączkowska, Agnieszka Brochocka, Aleksandra Nowak, Mateusz Wojtkiewicz

**Affiliations:** 1Laboratory of Respiratory Protective Equipment, Department of Personal Protective Equipment, Central Institute for Labour Protection—National Research Institute, Wierzbowa 48, 90-133 Łódź, Poland; 2Department of Molecular Physics, Faculty of Chemistry, Łódź University of Technology, Żeromskiego 116, 90-924 Łódź, Poland

**Keywords:** gas sensor, chemosensitive layers, resistance, ammonia vapors

## Abstract

The developed chemosensitive layers consisted of multi-walled carbon nanotubes (MWCNTs), reduced graphene oxide (rGO), and a conductive polymer (polyaniline—PANI) in a polymeric matrix (a polystyrene solution in methylene chloride). The layers were challenged with a test gas to determine the optimum variant in terms of sensitivity to the selected analyte and the repeatability of results. In terms of individual components, the greatest percentage change in resistance (32%) and the best repeatability were found for chemosensitive layers containing a PANI salt in the polymeric matrix. Even greater changes in resistance were exhibited by sensors containing more than one active component in the matrix: 45% for PANI + MWCNTs and 75% for PANI + rGO. The presented method of thin-layer deposition was shown to be suitable for the production of sensitive and functional sensors of ammonia vapors. The developed sensors were characterized by high repeatability and sensitivity to a harmful substance that constitutes an inhalation hazard to workers. The sensors were also analyzed for their durability and recovery as well as the ability to function under varying temperature and humidity conditions.

## 1. Introduction

Gas sensors are devices that convert a fraction of gas volume into an electric signal. They are of crucial importance in environmental monitoring, industrial chemical processing, public safety, agriculture, and medicine. With the progress of science and technology, efforts are under way to identify new gas-detecting materials to improve the sensitivity, selectivity, response time, cost effectiveness, energy efficiency, stability, and portability of gas sensors [1]. Some of these materials show great promise as indicators of the loss of the protective properties of personal protective equipment (PPE) and in particular respiratory protective devices (RPDs), which are often deployed in conditions of direct threat to human health and life.

Highly sensitive and specific gas sensors are used to decrease the likelihood of explosions by detecting leaks of flammable or explosive gases; they can also be used for the real-life detection of toxic or harmful gases in industry. In addition, portable electrochemical biosensors are used in agricultural, electrochemical, or food production technology, e.g., they are used to detect pesticides in fruits and vegetables [2].

Moreover, there is growing public awareness of the need for an improved capacity to monitor the quality of the atmosphere in the context of eliminating factors polluting the natural environment [3]. The basic criteria for effective gas sensors include high sensitivity and selectivity, fast response and recovery time, low power consumption, reliability, and measurement independent of changes in temperature and humidity (relative to detection sensitivity). The commonly used gas sensors contain gas-responsive polymers [4], as well as semiconducting metal oxides and materials characterized by high porosity, such as porous silicon. Gas sensors typically take advantage of changes in the physical/chemical properties of materials attributable to adsorbing or desorbing gas particles that are found in their immediate vicinity. Therefore, sensors are most often built from materials exhibiting high specific surface area, such as carbon nanoparticles (fullerene, graphene, single- and multi-walled carbon nanotubes, and carbon nanofibers) [5]. Due to their unique structure, as well as electrical, mechanical, thermal, and optical properties, of greatest interest among sensing materials are carbon nanotubes, which were discovered by Iijima in 1991 [6].

Nanotubes are among the strongest and most rigid known nanomaterials. Their smallest diameters are measured in nanometers, while their lengths can be up to millions of times greater. They are characterized by extraordinary resistance to rupture and unique electrical properties. Depending on the arrangement of bonds (longitudinal or transverse), nanotubes may act as conductors or semiconductors; theoretically they can conduct 1000 times more current than metal cables of the same weight. They are also excellent heat conductors and display considerable chemical stability as well as very high capacity for reversible elastic deformations under compressive and bending forces due to their flexibility and hollow structure. As a result of all these properties, carbon nanotubes can be applied as sensor materials for detecting organic substances that adsorb on porous solids, and especially on activated charcoal used as the main component of filters.

To date, carbon nanotubes have been shown to be sensitive to the presence of gases such as oxygen, hydrogen, nitrogen, water vapor, nitrogen dioxide, carbon oxide, sulfur dioxide, acetone, methanol, ammonia, chloroform, toluene, and benzene [7]. Sensors incorporating them have been reported to exhibit short response times and sensitivity exceeding that of commercially available semiconductor sensors. These discoveries have led to a rapidly growing interest in the application of carbon nanotubes for the detection of gases and hazardous vapors. Efforts have been made to design novel sensors as well as develop methods for the functionalization of sensor layers with a view to improving their sensitivity, response, recovery time, and selectivity, as well as reducing their costs and power consumption and streamlining production methods [8,9,10]. Researchers have proposed numerous ways of producing gas sensors incorporating highly ordered nanotube arrays and random lattices [11]. Due to their electrical properties, considerable interest has recently been shown in conducting polymers. These characteristics encourage scientists to continue research on their applications. One of the most promising conducting polymers is polyaniline (PANI) stable conductive form, wide analytical range, and potential technological uses in many areas. It is characterized by lightness, processability, corrosion resistance and a wide range of electronic conductivity, which can be achieved by doping. Its properties have been used extensively, e.g., to produce batteries, coaxial cables, displays, gas sensors, bio-sensors, electrostatic and conductive materials or even artificial muscles, as presented by the authors in their work [12]. The chemical structure of polyaniline, including its conjugated bond system, is a necessary but insufficient condition for obtaining a conductive structure. Porosity can be controlled by the doping system and certain organic solvents [13,14]. Lina Xue et al. developed a flexible electronic gas sensor using a composite consisting of PANI and carbon nanotubes (CNTs) for detecting ammonia vapors. Their sensor was characterized by high sensitivity to the vapors (200 ppb to 50 ppm at ambient temperature), a fast response (85 s), and a rapid recovery time (20 s). According to the authors, the good ammonia-sensing properties were probably attributable to the synergistic effects of the PANI/CNTs nanocomposite with a BET specific area of 54.187 m^2^/g [15]. In turn, Ching-Ting Lee developed sensors based on a composite incorporating PANI and reduced graphene oxide (rGO). The optimized sensitivity of their sensors was 13% for ammonia vapors at a concentration of 15 ppm at ambient temperature and a humidity close to 0%.

The sensitivity of PANI/rGO gas sensors was much higher than that of selective gas sensors containing either rGO or PANI, at 0.5% and 8.3%, respectively. Moreover, as the sensitivity of PANI/rGO sensors exhibited similar values at different relative humidity levels, the sensors can be used in very humid environments including gases exhaled by humans [16]. The adsorption properties of polyaniline were also used by Rima D. Alharthy and Ahmed Saleh in the work [17]. They showed that the PET-PANI and PET-PANI-CoFe_2_O_4_ sensors have the highest values of selectivity when exposed to NH_3_ gas at room temperature compared to other environmental gases. The PET-PANI-CoFe_2_O_4_ (50%) flexible sensor demonstrated a maximum response value of 118% and an excellent response time of 24.3 s at 50 ppm NH_3_ gas. The sensor has good reproducibility and an ultra-low detection limit.

Matsuguchi et al. developed a sensor based on PANI microspheres to detect NH3 vapors. They showed that many spaces are created inside the film by producing a PANI sensor layer uniformly covering micrometer-sized spheres. This enhances the reaction between NH3 and PANI molecules. They also showed that the sensor response increases with the diameter of the microspheres [18]. Other researchers used the MoTe2/PANI nanocomposite to fabricate an NH3 gas vapor sensor. They showed that the produced sensor has a faster and four times better response to ammonia in the range of 10–1000 ppm than the pure PANI sensor [19]. Polyaniline has also been used by Ciril Reiner-Rozman et al. to make a PANI film doped with 4.25% wt. F4TCNQ. It can be used to detect tumor markers, and their ammonia detection efficiency was 100 ppb, which is lower than in the case of sensors with PANI alone [20]. Studies have shown that doping polyaniline with various materials improves the detection properties.

The aim of the work was to develop chemically sensitive layers made of materials with a developed specific surface area for use in sensors for substances harmful and dangerous to human health and life and to analyze the electrical response of the sensor, taking into account the changing conditions of its use.

## 2. Materials and Methods

### 2.1. Materials

The dispersion systems used for the fabrication of chemosensitive layers detecting toxic and hazardous substances contained the following substances: methylene chloride (CH_2_Cl_2_, STANLAB, Lagos, State, NG), polystyrene (PS, Sigma Aldrich, St. Louis, MI, USA), polyaniline (PANI, Sigma Aldrich), multi-walled carbon nanotubes (MWCNTs, Ossila, Sheffield, UK), and reduced graphene oxide (rGO, University of Bielsko-Biała). The sensor consists of a chemosensitive layer in the form of a mixture of carbon nanotubes and polyaniline deposited by spraying using the developed automated air brush system. This system enabled the application of a chemically sensitive mixture in the form of 6 thin layers on the comb electrodes in a continuous reciprocating motion. The stream of the chemically sensitive mixture coming out of the nozzle was sprayed with compressed air at an angle of 90° from a height of 10 cm. The materials used in the sensors were tested by means of low-temperature nitrogen adsorption–desorption in order to elucidate their textural properties, i.e., Brunauer–Emmett–Teller (BET) surface area, total pore volume, and mean pore size. The obtained measurement results are given in Table 1. Images from an optical microscope presenting the morphology of films produced by drop-casting, the depth profile presenting the roughness of the layers and the role of the selected materials, and their features, as well as the reason for choosing them, were broadly discussed in the previous paper—Brochocka, A.; Nowak, A.; Zajączkowska, H.; Sieradzka, M. Chemosensitive Thin Films Active to Ammonia Vapours. *Sensors* 2021, 21, 2948. https://doi.org/10.3390/s21092948 (accessed on 22 April 2021) [11].

Prior to analysis, samples were degassed in vacuum at 200 °C for 4 h.

The polymeric matrix consisted of a PS solution in CH_2_Cl_2_. The optimized concentrations and qualitative compositions of the studied dispersion variants are given in Table 2.

### 2.2. Research Methods

#### Method of Sensor Production

The dispersion technology involved two mixing steps: one using an MS 11 magnetic stirrer (Wigo, Poland) and another one using a SONIC-2 ultrasonic bath (POLSONIC Palczyński Sp. J.). The substrates for the sensors were copper comb electrodes printed on a 1 mm-thick epoxy laminate and coated with gold. 

Each dispersion system variant was deposited on a substrate with comb electrodes by spraying using a developed automated air brush system. This made it possible to fabricate repeatable active layers with known and well-defined physicochemical parameters. Chemosensitive layers prepared in this way were dried in a vacuum dryer (BINDER GmbH, Tuttlingen, Germany) in order to eliminate residual solvents and water absorbed from the air during the deposition process.

### 2.3. Testing the Electric Response of the Fabricated Sensors

The electrical response of the sensors was tested on a station equipped with three rotameters supplying dried compressed air to the measurement system. Dräger X-am 7000 gas detectors (Drägerwerk AG & Co., KGaA, Lübeck, Germany) were used to measure the concentration of test vapors in the control and measurement chambers. Aerosol containing the test substance was generated in an evaporator, where the substance was exposed to dried compressed air. Subsequently, the vapors were fed into a stirrer, where they were volumetrically mixed with dried compressed air to form an air/test substance aerosol. Tests were conducted at a volumetric flow rate of 30 L/min and a constant temperature of (24.0 ± 0.6) °C and humidity (6.3 ± 1.4)%. Moreover, an artificial lung and a thermohygrometer were installed in the system to conduct tests at higher temperature and humidity levels: (26.0 ± 1.0) °C and (70 ± 10)%, (30.0 ± 1.0) °C and (80 ± 10), and (35.0 ± 0.6) °C and (90.0 ± 10.0)%. Air supply was set to 2 L/stroke at 25 strokes/min; the air was supplied to the measurement chamber through an artificial lung (a humidifier with a heater).

A digital laboratory multimeter (KEYSIGHT Technologies 34461A Digit Multimeter, Truevolt, Santa Rosa, CA, USA) was used to measure resistance and analyze changes in this parameter before and after sensor exposure to the analyte.

Based on measurements, time curves were plotted for the resistance of the chemosensitive layers deposited on comb electrodes exposed to the test aerosols. Sensor variants characterized by good sensitivity and response to the presence of the test aerosols were determined on the basis of percentage changes in resistance. The main goal was to obtain the highest percentage change in resistance (S) with respect to the baseline value for the selected test aerosol, as determined from the formula:S = ((R_max_ − R_0_)/R_0_) × 100% (1)
where: S—percentage change in resistance, R_max_—maximum resistance after sensor exposure to aerosol, and R_0_—baseline resistance (for sensors exposed to pure air).

## 3. Results and Discussion

The developed sensor variants were exposed to test aerosol flow to measure their electrical responses. In most tests involving ammonia, the concentration of its vapors was close to the maximum allowable concentration (MAC) of approx. 18.7 ppm. However, in one experiment analyzing the effects of ammonia concentration, it was set to values either lower or higher than the MAC: 6 ppm, 10 ppm, 26 ppm, and 30 ppm. Table 3 contains the highest changes in the resistance of the developed chemosensitive layers, while Figure 1 shows the electrical response of those layers to ammonia vapors.

Sensors incorporating MWCNTs as the only active component exhibited low changes in resistance. The obtained resistance curve for ammonia (Figure 1a) was consistent with expectations: in the first step of the experiment, sensor exposure to pure air resulted in a constant value of resistance; in the second step, resistance rose as ammonia passed through the test chamber; in the third step, resistance decreased and stabilized after analyte flow was terminated and pure air was passed through the chamber. These changes in resistance reflect desirable, fast sensor responses. This is consistent with the results reported by Lina Xue and collaborators, who developed PANI sensors characterized by short response times.

Sensors incorporating rGO exhibited resistance changes of approx. 5.79%, which were much greater than those reported by Ching-Ting Lee for their selective rGO sensor. The resistance changes obtained in the present work were well-defined and rapid. Resistance changes induced by the test aerosol in the rGO sensor are shown in the figure. The response of the rGO sensor was opposite to that of the MWCNTs sensor: in this case resistance decreased under the influence of ammonia, to rebound after exposure to pure air. This behavior is associated with a different mechanism of interaction between the test aerosol and the chemosensitive layer. This observation was already made by us in the previous paper: Brochocka, A.; Nowak, A.; Zajączkowska, H.; Sieradzka, M. Chemosensitive Thin Films Active to Ammonia Vapours. *Sensors* 2021, 21, 2948. https://doi.org/10.3390/s21092948 (accessed on 22 April 2021) [11]. PANI sensors revealed the highest resistance change among all the developed single-component sensors. The resistance time curve for ammonia was consistent with expectations and exhibited very high dynamics and negligible noise.

Sensors incorporating both MWCNTs and PANI displayed even higher resistance change than those containing PANI alone. In addition, similarly as in the case of PANI sensors, the resistance time curve for ammonia vapors was consistent with expectations and also showed very dynamic changes and negligible noise levels.

Sensors incorporating rGO and PANI revealed the highest resistance change values among all the studied variants. Similarly as in the case of the other sensors containing PANI, the resistance time curve for ammonia vapors was consistent with expectations and reflected a dynamic response and negligible noise. 

The results show that the application of a mixture of PANI and one of allotropic forms of carbon (rGO or MWCNTs) improves the detecting capacity of the sensors by increasing the maximum change of resistance following exposure to ammonia vapors.

The obtained MWCNTs + PANI and rGO + PANI gas sensors exhibited much higher changes in resistance in response to ammonia vapors than those proposed by Lina Xue et al. but also by Ching-Ting Lee and collaborators [13,14]. This may be due to the fact that the properties of the two components combined: MWCNT or rGO with a high specific surface area and good adsorption properties of polyaniline.

Due to the promising results obtained for sensors incorporating PANI and MWCNTs + PANI, in the next step, we analyzed the effects of different concentrations of the analyte that may occur in workplaces, rather than only the MAC level. For this purpose, we selected two concentrations of ammonia vapors below the MAC (18.7 ppm) and two concentrations above it.

Despite the fact that the greatest changes in resistance were obtained for the variant incorporating rGO and PANI in a PS matrix, further tests were conducted using sensors containing MWCNTs and PANI, as the latter combination displayed better repeatability of results (stability of resistance in air and of the maximum resistance when exposed to the analyte).

To determine the sensitivity of the tested sensors, they were challenged with known concentrations of the test aerosol. Ammonia molecules were found to induce changes in the resistance of the chemosensitive layer, as shown in Table 4. The exposure of the sensor to the analyte caused dramatic changes in its resistance. Even a very low concentration of ammonia vapors (6 ppm) increased the baseline resistance of the chemosensitive layer by 10% in the case of the PANI variant and by approx. 20% in the case of the MWCNTs + PANI variant. A further increase in ammonia concentration led to a proportional rise in the resistance of the chemosensitive layer to 40% of the baseline value for 30 ppm. At the same time, the experimental results were shown to be highly repeatable.

One of the most important parameters determining the sensor applicability for a specific purpose is its performance under conditions of cyclic changes in its working environment. Figure 2 shows the results of fatigue tests for PANI + PS and MWCNTs + PANI + PS sensors exposed to ammonia vapors.

In fatigue tests, the chemosensitive layer was exposed to cyclic ammonia aerosol flow followed by pure air. As can be noted, in both cases the resistance of sensors increased rapidly when exposed to the analyte. After terminating the flow of ammonia vapors, resistance decreased, although in each cycle the value of the decrease was much lower than the preceding increase, which led to an overall growth of sensor resistance irrespective of the number and duration of fatigue cycles. The incorporation of MWCNTs significantly affected the electrical properties of the detecting layer, tripling its sensitivity to ammonia vapors compared to layers not containing MWCNTs. Under the conditions of the carried-out test, the return to the initial value of the resistance was impossible due to the absorption of moisture and oxygen by the active layer, as well as residual amounts of analyte in the test chamber when purging it with clean air. It is highly probable that, when the sensor would be treated by annealing in vacuum conditions in order to get rid of adsorbed water, oxygen, and analyte molecules from the active layer, the sensor resistance would return to the initial value.

Another important area of research was determining the parameters of the chemosensitive layers under conditions of thermal and humidity stress. Tests were conducted to elucidate the effects of external conditions on the electrical parameters of those layers, as presented in Figure 3, Figure 4, Figure 5 and Figure 6.

It was found that sensors exposed to high temperature and humidity responded with a rapid drop in resistance for all samples irrespective of the composition of the chemosensitive layer, temperature, and the presence of ammonia vapors. For both examined layers, this was followed by resistance stabilization at temperatures up to 30 °C and humidity up to 80%.

However, the response of sensors exposed to extreme conditions of very high temperature and humidity (90%) was quite different, as shown in Figure 7 and Figure 8.

In these cases, a rapid drop in resistance was followed by a sudden increase in that parameter. The rate of resistance increase was proportional to the humidity of the medium to which the system was exposed. 

The resistance of sensors exposed to ammonia vapors decreased rapidly, similarly as in the case of those exposed to pure air, after which the trend was reversed and resistance rose much above the baseline value. This means that the sensitivity of the sensor to ammonia vapors was not affected by varying external conditions.

## 4. Conclusions

The developed sensor for detecting ammonia vapors PANI/MWCNTs is designed for applications in filtering respiratory protective devices. It can be used for determining the end of protection offered by those devices by measuring the concentration of ammonia vapors downstream of the adsorption bed. The sensor is characterized by high fatigue resistance as well as repeatable cyclic operation, which opens up new opportunities for industrial applications. The response time of the developed sensor depends on the thickness and homogeneity of the active chemically sensitive layer applied to the sensor and the sorption capacity of the absorber used. The developed PANI/MWCNTs sensors can be used many times. The influence of moisture causes a decrease in the resistance value of the sensor, but this does not affect its sensitivity to ammonia vapors. The fact that further usage does not require complete recovery is a great advantage; saves energy, resources, and money; and is very beneficial in its practical applications. Irrespective of ammonia concentration, detection is rapid and well-defined. This is a very important feature of the sensor, which makes it possible to deploy it under the actual conditions of use for respiratory protective devices and on an industrial scale.

Currently, there is no sensor on the market that absorbs inorganic substances with a low threshold of odor detection in real time, implemented in absorbing respiratory protective equipment and commercially available. Emission of vapors and gases of various inorganic substances is associated with many technological processes. Due to their toxic and harmful properties, they are the greatest threat in the working environment, both for the health and life of employees. Therefore, a sensor was developed to monitor and detect ammonia vapor, which informs the user when the absorbent element need to be replaced with a new one. The developed sensor is very sensitive and reacts faster to ammonia vapors already at 1 ppm than a gas analyzer. 

Thanks to the use of individual respiratory protective devices in the form of absorbers with a sensor sensitive to ammonia vapors, the risk of harmful substances affecting the health of employees will be reduced.

## Figures and Tables

**Figure 1 polymers-15-00420-f001:**
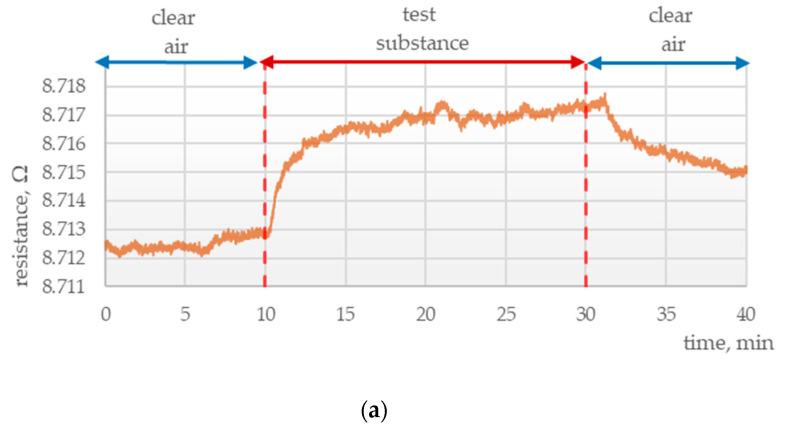
Electrical responses of sensors incorporating (**a**) MWCNTs, (**b**) rGO, (**c**) PANI, (**d**) rGO + PANI, (**e**) MWCNTs + PANI to ammonia vapors at a constant temperature of (24.0 ± 0.6) °C and humidity (6.3 ± 1.4)%.

**Figure 2 polymers-15-00420-f002:**
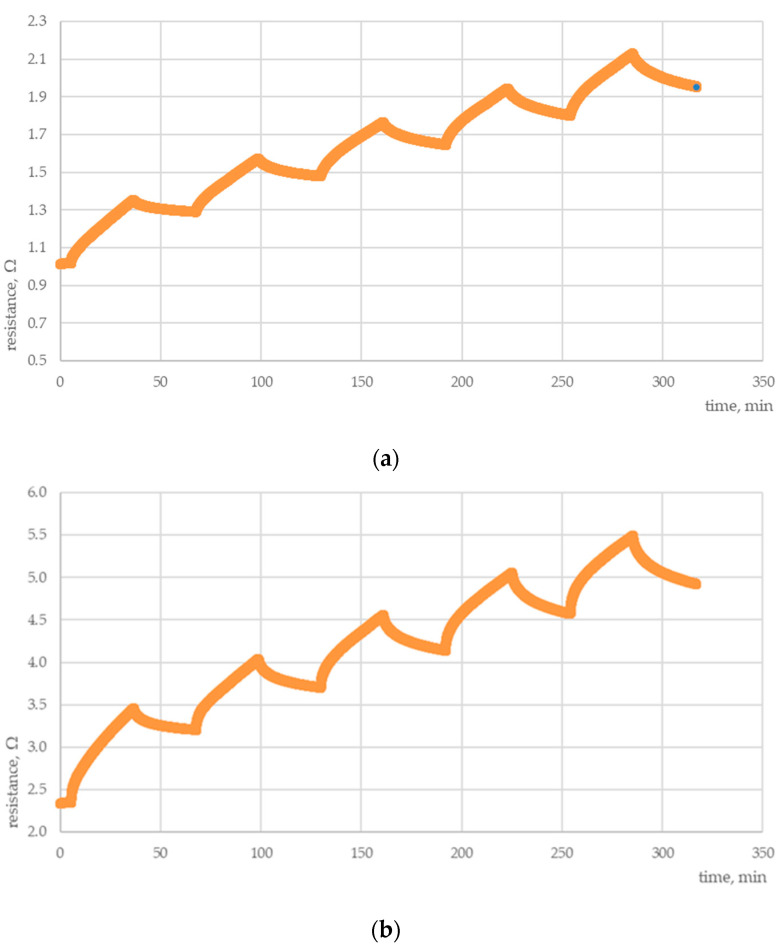
Electrical response of sensors incorporating (**a**) PANI + PS and (**b**) MWCNTs + PANI + PS in fatigue tests for exposure to ammonia vapors.

**Figure 3 polymers-15-00420-f003:**
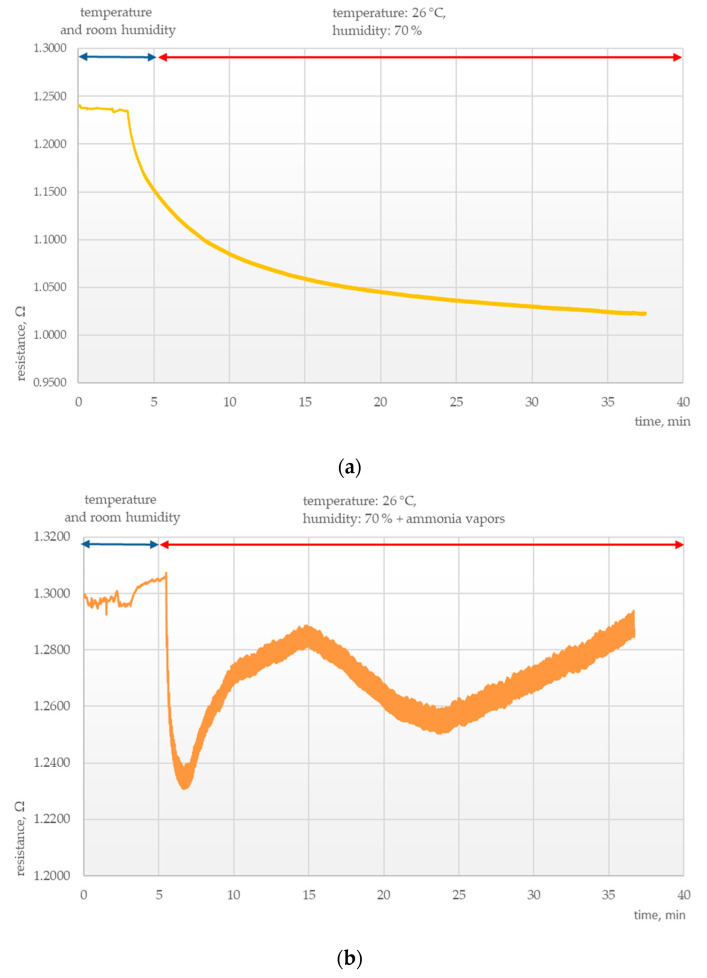
Electrical response of a PANI sensor at 26 °C and 70% humidity (**a**) without ammonia vapors and (**b**) with ammonia vapors.

**Figure 4 polymers-15-00420-f004:**
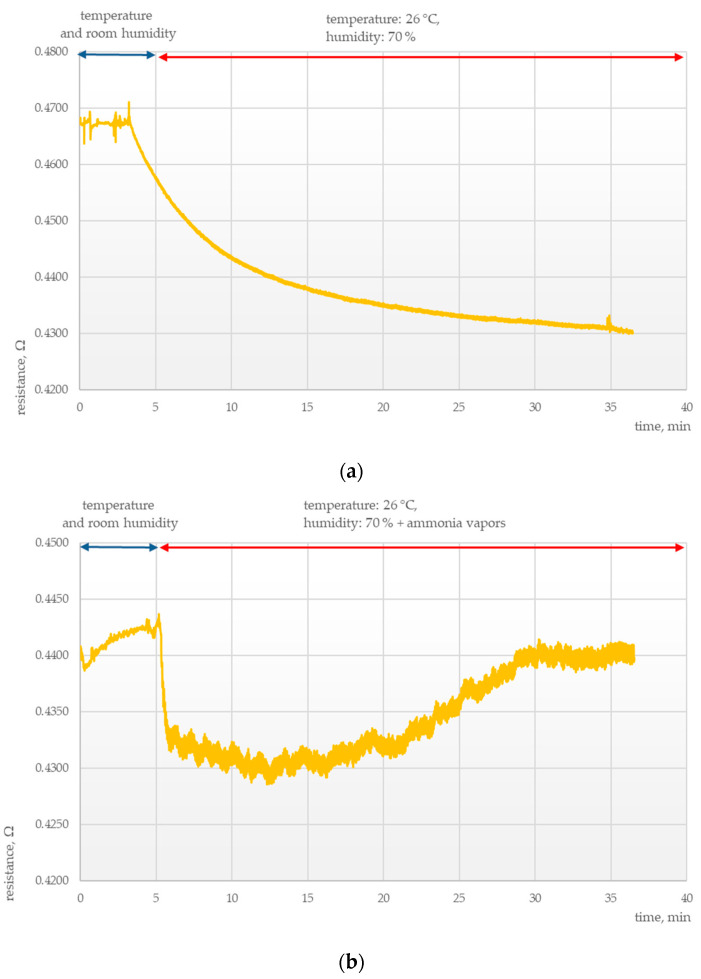
Electrical response of a MWCNTs + PANI sensor at 26 °C and 70% humidity (**a**) without ammonia vapors and (**b**) with ammonia vapors.

**Figure 5 polymers-15-00420-f005:**
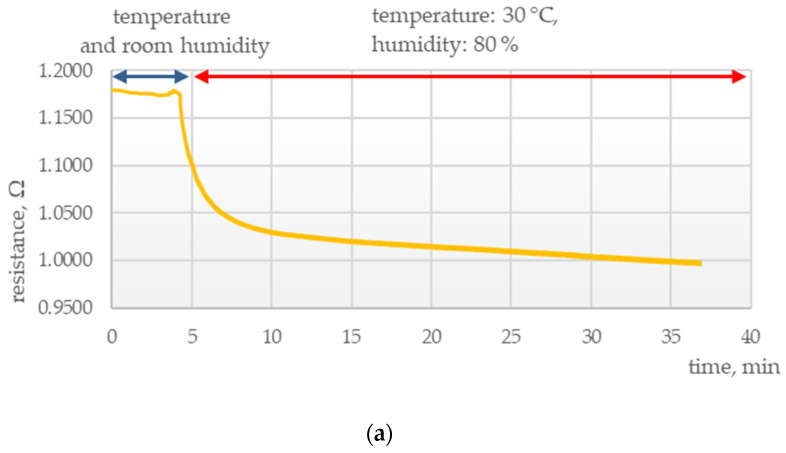
Electrical response of a PANI sensor at 30 °C and 80% humidity (**a**) without ammonia vapors and (**b**) with ammonia vapors.

**Figure 6 polymers-15-00420-f006:**
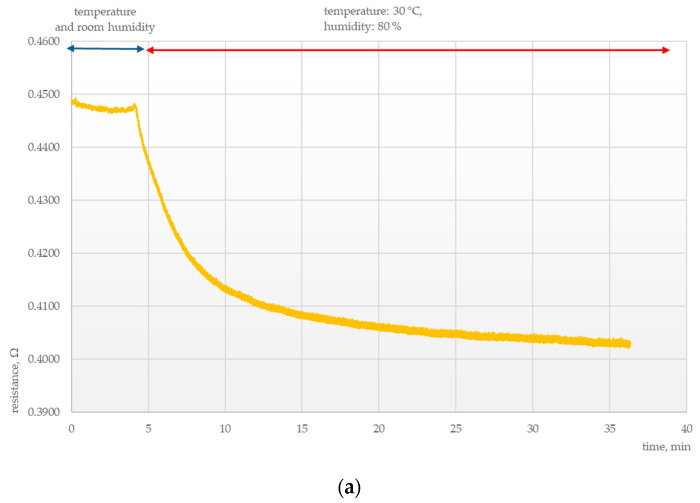
Electrical response of a MWCNT + PANI sensor at 30 °C and 80% humidity (**a**) without ammonia vapors and (**b**) with ammonia vapors.

**Figure 7 polymers-15-00420-f007:**
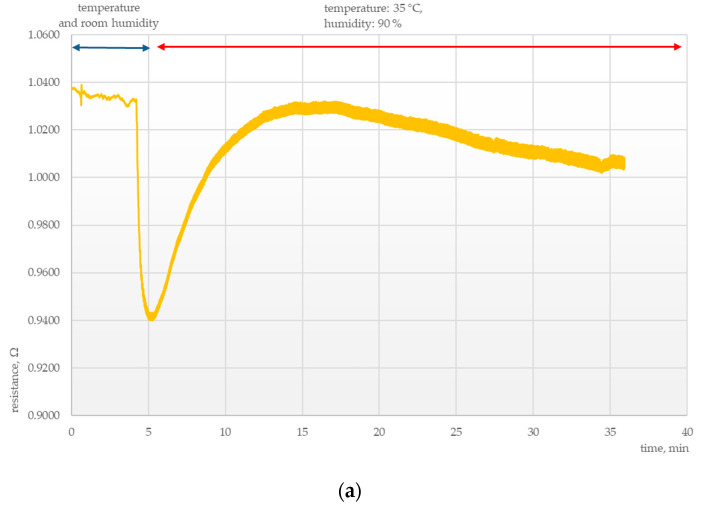
Electrical response of a PANI sensor at 35 °C and 90% humidity (**a**) without ammonia vapors and (**b**) with ammonia vapors.

**Figure 8 polymers-15-00420-f008:**
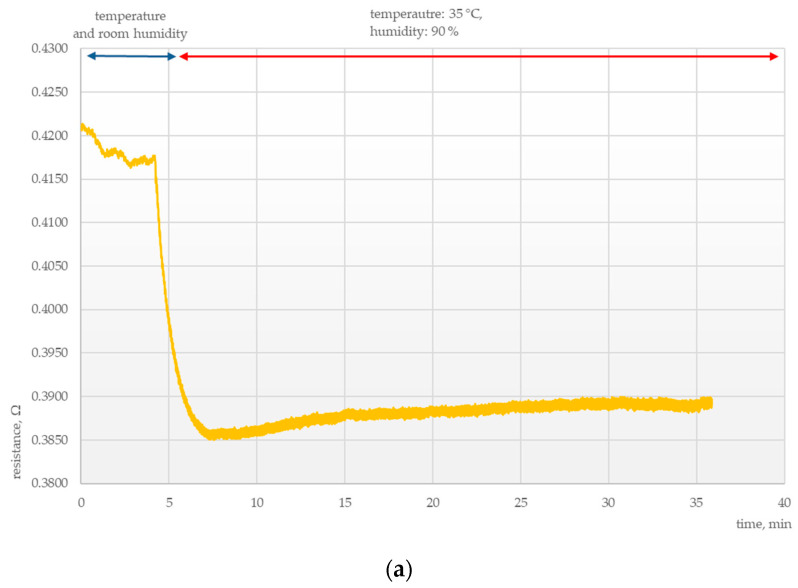
Electrical response of a MWCNTs + PANI sensor at 35 °C and 90% humidity (**a**) without ammonia vapors and (**b**) with ammonia vapors.

**Table 1 polymers-15-00420-t001:** Low-temperature nitrogen adsorption/desorption (BET) results.

No.	Sample	S_BET,_ m^2^/g	V_pore_, cm^3^/g	r_pore_, nm
1.	PANI	2.84	0.0222	21.7
2.	rGO	386	2.08	3.37
3.	MWCNTs	324	2.22	11.3

S_BET_—specific surface area calculated from the Brunauer–Emmett–Teller equation; V_pore_—total pore volume; r_pore_—mean pore radius.

**Table 2 polymers-15-00420-t002:** Quantitative and qualitative compositions of dispersion systems.

No.	Polymeric Matrix		Amount of Modifier, mg		
	CH_2_Cl_2_, cm^3^	PS, g	MWCNTs	rGO	PANI
1.	70	1	50	-	-
2.			-	50	-
3.			-	-	1000
4.			50	-	1000
5.			-	50	1000

**Table 3 polymers-15-00420-t003:** Highest changes in the resistance of the developed chemosensitive layers.

	MWCNTs	rGO	PANI	MWCNTs + PANI	rGO + PANI
**S**	0.06%	5.79%	31.66%	45.79%	75.72%

**Table 4 polymers-15-00420-t004:** Percentage changes in the resistance of chemosensitive layers of sensor as a result of exposure to different ammonia vapor concentrations.

	Mean Resistance Change after 30 min. for Different Variants of Sensor Formation, %	
Ammonia Vapors Concetration, ppm	PANI + PS	MWCNTs + PANI + PS
6	10	20
10	15	26
26	33	40
30	40	43

## Data Availability

The data presented in this study are available on request from the corresponding author.

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
