# Peer review of "Analysis of the Resistance Change of Chemosensitive Layers to the Presence of Ammonia Vapors under Variable Conditions of Air Temperature and Humidity"

_polymers, 2023, doi:10.3390/polym15020420_

Round 1

Reviewer 1 Report (Previous Reviewer 3)

The authors justified the reviewer's comments. The revised manuscript can be accepted for publication

Author Response

Thank you very much for your comments

Best Regards

Agnieszka Brochocka

Reviewer 2 Report (Previous Reviewer 2)

The response to the comment is satisfied. 

Author Response

Thank you very much for your comments

Best Regards

Agnieszka Brochocka

Reviewer 3 Report (New Reviewer)

Manuscript ID: polymers-2133125

Title: Analysis of the resistance change of chemosensitive layers to the presence of ammonia vapors under variable conditions of air temperature and humidity

Authors: Hanna ZajÄ…czkowska, Agnieszka Brochocka *, Aleksandra Nowak, Mateusz Wojtkiewicz

Authors are required to address these comments for improvement of the paper.

·         In line numbers 102 to 105, 115, 138. While writing chemical formulas, follow the correct method. Example: Subscripts are commonly used in chemical formulas. A scientist would write the formula for water, H2O, so that the 2 appears lower and smaller than the letters on either side of it.

·         Include a graphic representation of the possible mechanism for chemosensitive ammonia sensing.

·         Replace the faded figures and very light colours in the X and Y axis with clear and good ones.

·         What is the reason for so much difference in resistance observed in Figure 1 (a) MWCNTs, (b) rGO, (c) PANI, (d) rGO+PANI, (e) MWCNTs+PANI explain it?

·         What is the humidity effect observed on PANI composites?

·         Conclusions must be precise.

·         The authors who have carried out the present research must be appreciated. Since authors have to refer to recent appropriate articles of publication, here are a few more published articles that may be included.

doi.org/10.3390/s22145379

doi.org/10.1007/978-3-319-68255-6_186

doi.org/10.3390/chemosensors10070264

doi.org/10.3390/s21227522 

Author Response

The response to the reviewer's comments can be found in the attached file Reviewer 3 _EN1

Round 2

Reviewer 3 Report (New Reviewer)

The revised manuscript can be accepted. 

This manuscript is a resubmission of an earlier submission. The following is a list of the peer review reports and author responses from that submission.

Round 1

Reviewer 1 Report

In this work multi-walled carbon nanotubes (MWCNTs), reduced graphene oxide (rGO), and a conductive polymer (polyaniline – PANI) has been used to sense ammonia. It is reported that the greatest percentage change in resistance (32%) and the best repeatability were found for chemosensitive layers containing a PANI salt in the polymeric matrix. Even greater changes in resistance were exhibited by sensors containing 18 more than one active component in the matrix: 45% for PANI + MWCNTs and 75% for PANI + rGO. Some comments are highlighted below.

The objective/novelty of the work is unclear.

The title is very generic and does not give much information.

The surface morphology and crystal structure of the film does affect the sensing performance, however there are not such a investigation (SEM, AFM, XRD, FTIR, RAMAN etc).

The sensor is not fully recoverable, once the target gas is removed, how can such a sensor can be used for practical applications.

In the introduction section the authors highlighted that selectivity is an important factor for sensors, however the authors have not investigated the selectivity towards other gases such as hydrogen, hydrazine, nitrogen dioxide, acetone, methane and VOCs.

The conclusion of the paper is also missing, I think the authors mixed up discussions and conclusion sections.

Reviewer 2 Report

The manuscript presents chemosensitive layers consisted of multi-walled carbon nanotubes (MWCNTs), reduced graphene oxide (rGO), and a conductive polymer (polyaniline – PANI) in a polymeric matrix (a polystyrene solution in methylene chloride). Analysis of the electrical response of the sensors to the presence of ammonia vapors under variable conditions. There are some comments toward the research.

1.     The fonts of the in the figures (Figure 1, 2, 3, 4, 5, 6, 7 and 8) are too small to read. It is suggested to replot them.

2.     According to the definition of Sensitivity S=((Rmax-Ro)/Ro)x100%, Table 3 shows the rGO+PANI is 75.72%, but in Figure 1 (e) the Ro=0.6 W, Rmax=0.87 W, then the S will be 36.67%, that is the value of sensitivity in Table 3 is inconsistent to that of Figure 1. Similar error also found in Figure 1 (d).

3.     The response of rGO sensor was opposite to that of the MWCNYs sensor. The author claim it is associate with a different mechanism. It is suggested to provide more information about the different mechanism.

4.     Line 184, the author claims “PANI sensor revealed the highest resistance among all the developed single-component sensors.” But in Figure 1 (a) the baseline resistance of MWCNTs is about 8.712 W, Figure 1 (b) the baseline resistance of rGO is 0.327 MW. These values are higher than that of PANI.

5.     Line 191, the author claims “Sensor incorporating rGO and PANI revealed the highest resistance values among all the studied variants” It seems wrong as compared Figure 1 (d) and (e).

6.     Line 198-202, the high response of MWCNTs+PANI and rGO+PANI is due to the BET surface areas. Could you show the BET data of any reference?

7.     In Figure 2(a) and (b) why the response of the sensor did not recover to their original resistance values as the ammonia vapor is off?

Reviewer 3 Report

Current manuscript entitled “Analysis of the electrical response of novel sensors to the presence of ammonia vapors under variable conditions of their use” by “ZajÄ…czkowska et al” deliberated on the chemosensitive layers and their sensing behavior.The manuscript seems good and can be accepted after addressing the following comments.

1.      Authors should discuss about the role of the selected materials (MWCNTs; rGO and PANI)

2.      In the materials section provide detailed information on the “thin layer deposition”.

3.      Introduction last paragraph needs to be restructured; authors should mention what they have performed in the current work.

4.      Discussion section needs to be strengthened; discuss about your obtained results.

5.      There were no conclusions of the work.

6.      Some potential literature is dedicated on the on-site sensors for toxic pollutants and hazardous constituents’ detection that can provide a basic idea; those must be cited.

https://doi.org/10.1016/j.ccr.2021.214305

Polymers 2021, 13(18), 3077; https://doi.org/10.3390/polym13183077